# Gravelly Beach Deposits as a Proxy for Relative Sea-Level Changes in Microtidal Wave-Dominated Shoreline Systems: Examples from the Hinterland of the Taranto Gulf (Middle Pleistocene, Basilicata, Southern Italy)

Vincenzo De Giorgio *, Luisa Sabato and Marcello Tropeano

Dipartimento di Scienze della Terra e Geoambientali, Università degli Studi di Bari Aldo Moro, 70124 Bari, Italy;
luisa.sabato@uniba.it (L.S.); marcello.tropeano@uniba.it (M.T.)
* Correspondence: vincenzo.degiorgio@uniba.it; Tel.: +39-33-3670-3898

**Abstract:** The hinterland of the Taranto Gulf in Basilicata (Southern Italy) provides a great opportunity for the study of coarse-grained coastal systems belonging to a staircase of Quaternary terraced marine-deposits. Among gravelly successions, beach deposits abound in the stratigraphic record, offering exceptional outcrops useful for providing detailed information on their facies features. In this paper, we describe sedimentary facies, textural variations, and the depositional architecture of these deposits in order to: (1) demonstrate that the area is an excellent training ground for the study of gravelly beaches in microtidal settings; (2) discuss the use of beach deposits as a proxy for even small relative sea-level variations.

**Keywords:** gravelly beach deposits; relative sea-level changes; Metaponto coastal plain



## 1. Introduction

The hinterland of the Taranto Gulf in Basilicata, i.e., the Metaponto coastal plain and its inland (Southern Italy) (Figure 1), is characterized by the occurrence of coarse-grained coastal deposits located in the uppermost part of the exposed successions; the latter have been interpreted to be linked to a flight of marine terraces developed during the Quaternary because of the interference between regional uplift and eustatic sea-level changes [1]. Despite the large number of geomorphological studies of either regional or local significance and the production of several local chronostratigraphic data sets, authors working in the area disagree with each other about the age and distribution of these marine terraces. Moreover, a detailed sedimentologic study locally performed in the area demonstrated that each terrace-surface cannot be simply linked to a cycle of relative sea-level change and that coastal or alluvial coarse-grained deposits located in the uppermost part of the section, just below a terraced surface, could not be genetically related to some of the underlying (possibly datable) sandy deposits, whose sedimentation could be linked to a previous (older) and different relative sea-level/base-level position [2].

Since photointerpretation seems not to be useful to distinguish how many different marine-terraced surfaces developed in the area, studies exclusively based on geomorphologic methods are unable to describe the complex interaction between regional uplift and sea-level changes. Therefore, the age of samples not well constrained to a detailed measured and sedimentologic-interpreted succession cannot be simply and confidently used to relatively date the top of the sampled terraced succession. This means that it is necessary to perform a detailed facies analysis of the selected succession to place the samples both in the right paleoenvironment (i.e., its hypothetical paleodepth) and in the local history of relative sea-level changes before to dating outcropping deposits.

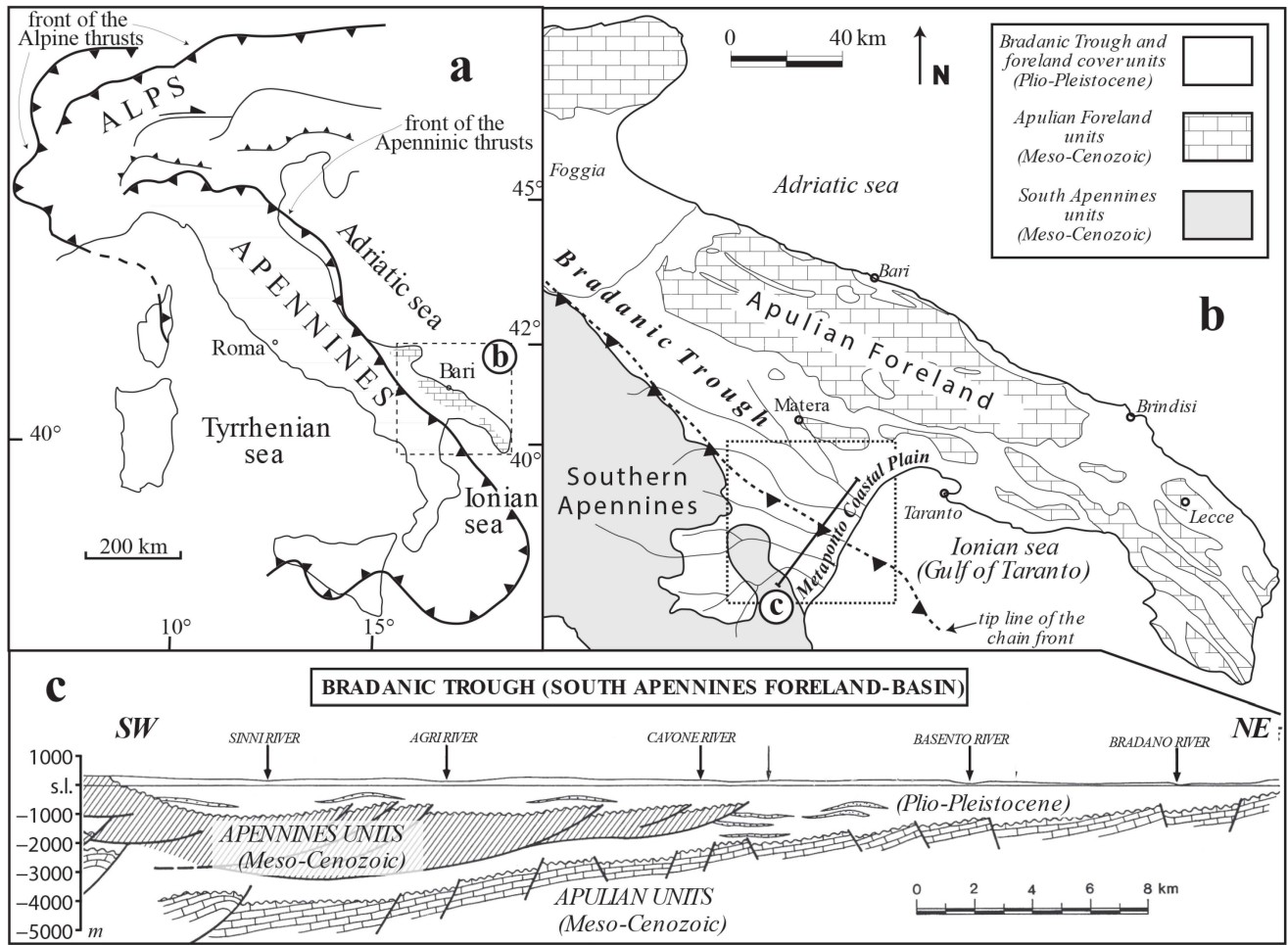

**Figure 1.** (**a**) Schematic structural map of Italy; (**b**) schematic structural map of Southern Italy; the dotted box indicates the geographical position of the next figure; (**c**) geological cross-section across the Bradanic Trough (after [3] modified). After [4], modified.

In clastic depositional coastal settings, like those ones whose sediments are exposed in the study area, the most sensitive environment to even the low-amplitude sea-level oscillations is the beachface, which, during its development, represents one of the best indicators (with a small error range) of the relative paleo sea level. In order to develop a future age-dating of uplifted Quaternary successions in the hinterland of the Taranto Gulf, the recognition of beachface facies (and adjacent ones), even those not directly linked to a terraced surface, represents one of the main tools to constrain deposits to the relative sea level to which they were genetically related.

After this premise, this work has a double purpose:

(1)   demonstrate that the area is an excellent training ground for the sedimentologic study of gravelly beaches in microtidal settings;

(2)   demonstrate that the sedimentologic approach must be the basis for following studies about the complex evolution of vertically stacked multistory paralic deposits topped by a terraced surface, considering that beachface deposits (and genetically adjacent ones) represent a proxy of the relative sea-level variation in the analyzed successions.

## 2. Geological Setting

The study area is located close to the town of Bernalda, in the hinterland of the Taranto Gulf (Basilicata, Southern Italy) (Figure 2). It corresponds to the Southernmost sector of the Bradanic Trough (the Southern Apennines foredeep) (Figure 1b), where Middle and Upper Pleistocene terraced marine-deposits developed and crop out due to a regional

uplift [5,6]. Geodynamic causes of uplift are still debated, being alternatively attributed to: (i) an isostatic rebound [7] induced by a slab detachment [8] (after [9]); (ii) a lithospheric buckling [10]; (iii) the combined activity of an out-of-sequence thrust sliding along the basal detachment of the external Apennines wedge and a lithospheric-scale duplexing [11]. The uplift started at least in the late Early Pleistocene [5,7,12] and the calculated uplift ranges between 0.4 and 1 mm/y [8,13,14] up to 2 mm/y [11]. The deepening of the drainage network, induced by uplift, led to the exposure of the upper part of the basin-fill deposits of the Bradanic Trough, i.e., a regressive succession made up of offshore silty clay deposits (Argille subappennine Fm) overlain by sandy and gravelly coastal deposits (marine terraces in Figure 2). These coarse-grained coastal deposits become younger and younger, moving from NW to SE, down to the Metaponto coastal plain, and are mainly represented by either progradational beaches or Gilbert-type deltas [12,15–18].

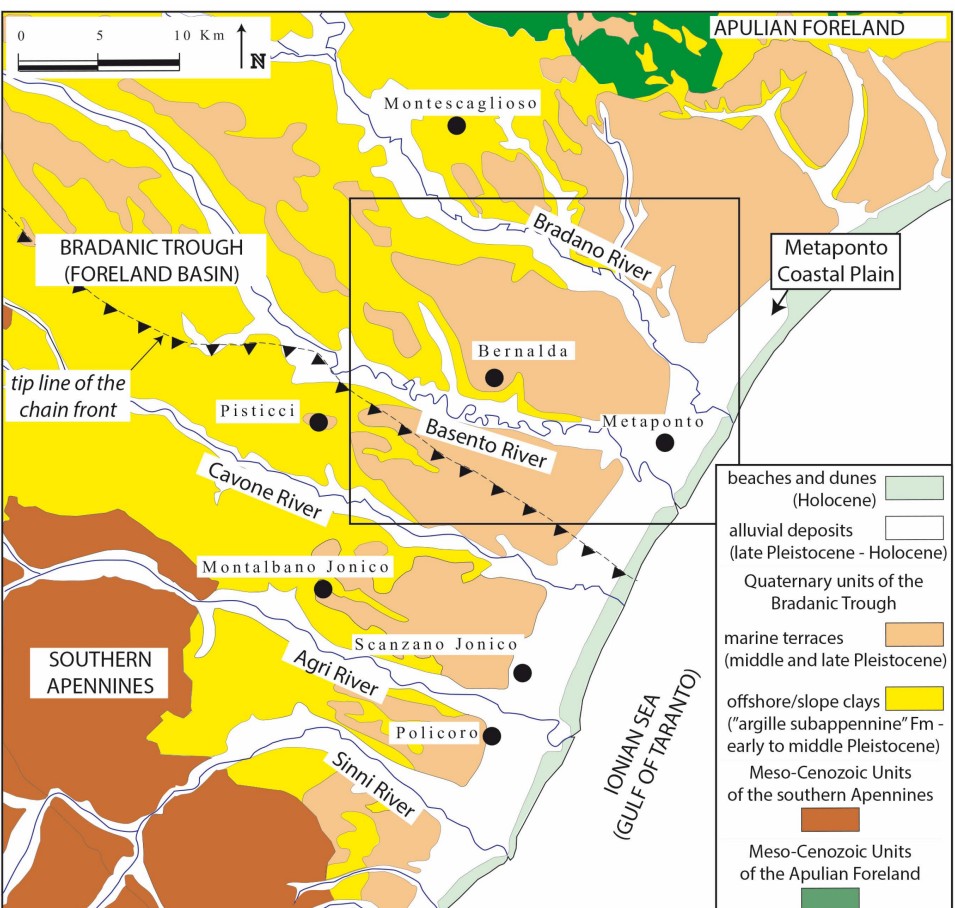

**Figure 2.** Schematic geological map of the region comprising the study area (see Figure 1 for location), located between the Bradano and Basento Rivers; the box indicates the geographical position of the next figure. After [19–21] (modified).

Except for [22], all the works regarding the area interpreted the terraced surfaces as related to a series of past relative sea-level high-stands. Regrettably, the number of surfaces and, consequently, the age of underlying deposits differ from author to author, and the proposed number, exclusively based on the identification of terraced surfaces, spans from 7 [15,16] to 11 [23,24], up to 18 [11]. As regards the study area, the recognized surfaces vary in number from 5 to 7 (Figure 3), but never was the genetical link of each surface to the "anatomy" of the underlying succession proposed, assuming that each surface represents the top of a single transgressive–regressive coastal cycle.

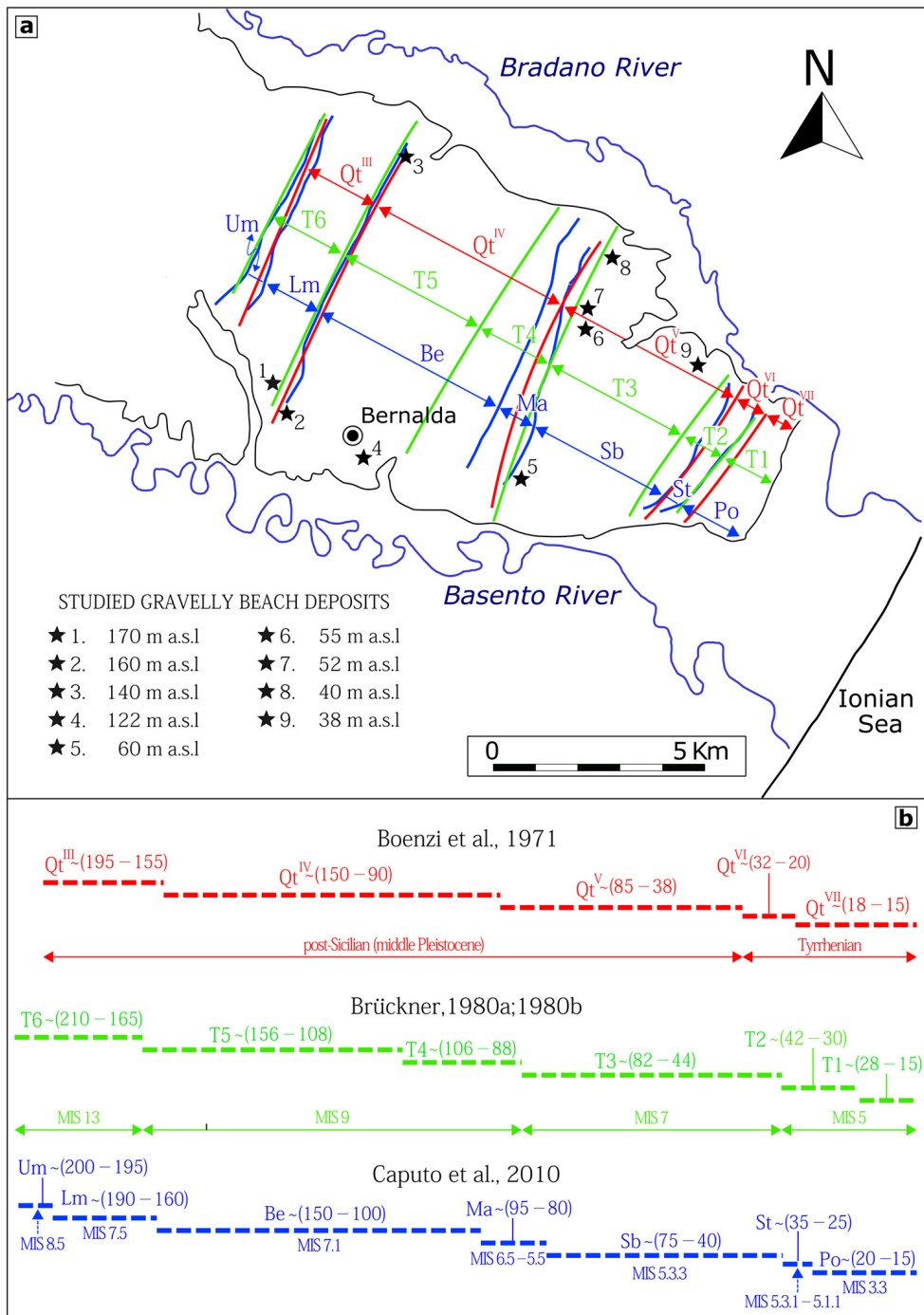

**Figure 3.** (**a**) Different interpretations of the number and distribution (arrows) of marine terraces in the study area (see Figure 2 for location), according to [16] (Qt^III–Qt^VII—boundaries in red), to [23,24] (T1–T6—boundaries in green), and [11] (Um–Po—boundaries in blue). The location of the studied beach successions is reported. (**b**) Different altitudinal distribution and age of marine terraces according to the same authors. MIS is the acronym for Marine Isotope Stage.

By contrast, and without entering in the definition of how many terraced surfaces could have been detected in the area, a detailed sedimentologic and stratigraphic study performed between the Cavone and Basento Rivers (adjacent to the area of the present study—see Figure 2 for the location of quoted rivers) highlighted the presence of different and very complex successions below the topographic surface (Figure 4). In order to avoid confusion, ref. [2] suggested the use of the term "terraced marine-deposits" to indicate these deposits are mainly marine in origin and terraced on top, but without affirming a

genetic relationship between the whole local coarse-grained succession and the flat surface above. The authors concluded that these terraced marine-deposits record more than a single relative sea-level change in the succession located below each terraced surface, even in the subsurface of the present-day coastal plain; therefore, terraced surfaces topping these deposits could not be genetically linked to the whole succession located below each of these surfaces but, eventually, they could be related to the last of those cycles vertically recorded in the local sedimentary succession [2,4,25,26]. Moreover, the "imbrication" of coastal wedges of different ages, i.e., a series of coastal bodies that cyclically developed one in front of the other at a different time, is worldwide documented and is characterized on top by an apparently single-terraced surface (a composite-terraced surface) [27–30].

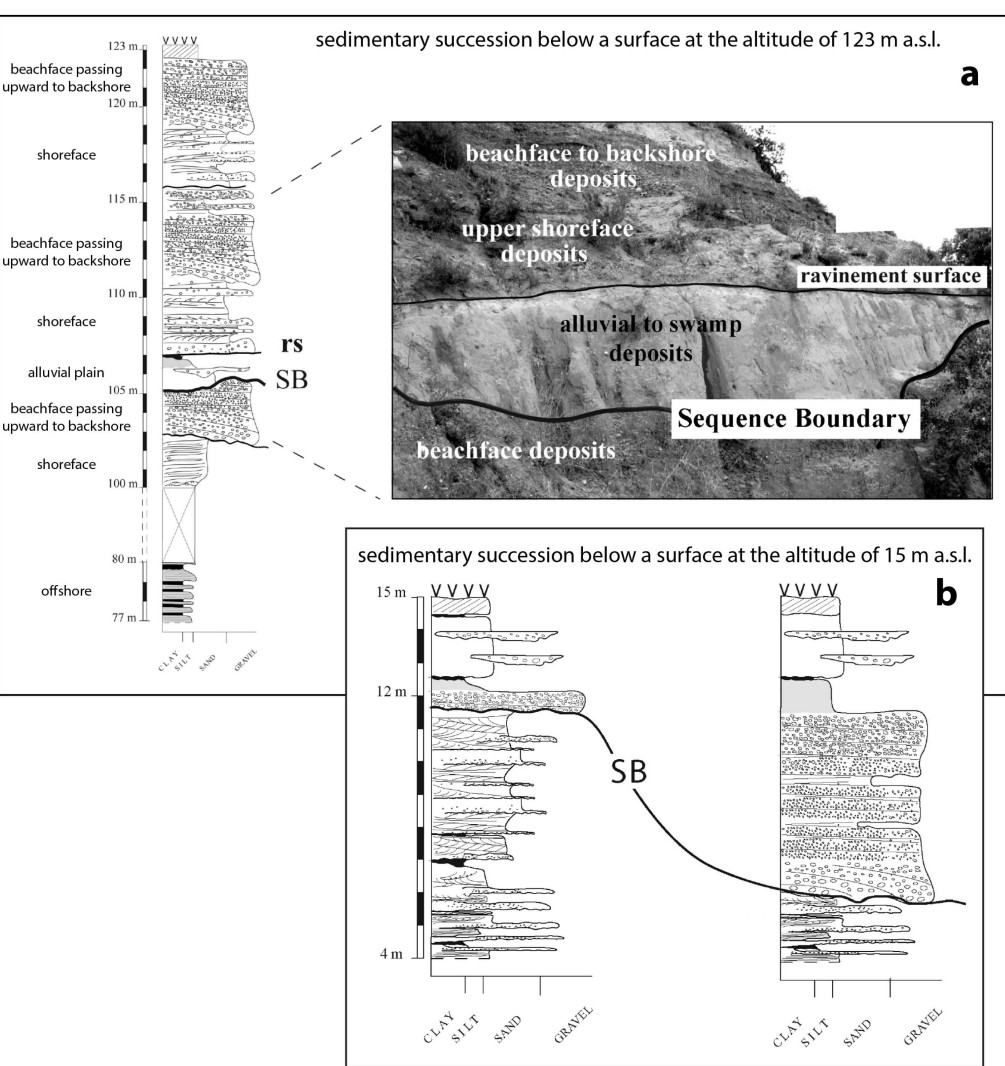

**Figure 4.** Two marine-terraced successions of the hinterland of Taranto Gulf were measured in detail by [2] in the vicinity of the area of the present study. In both cases, sampling without a reference log could lead to an incorrect age attribution to the terraced surface. (**a**) A log showing the stacking of 3 beach sequences (from shoreface to backshore) and the presence of alluvial deposits encased in marine ones. At least 3 cycles of relative sea-level changes are vertically recorded. (**b**) From the correlation of 2 sections close to each other, the presence of a beach sequence erosively lying (and pinching) on shoreface deposits was observed. The stratigraphic architecture suggests that shoreface deposits are related to a previous position of the relative sea level that dropped and led to the formation of an erosive surface. Deposition of gravelly beach deposits took place during the following relative sea-level rise and high-stand.

## 3. Methods

The present paper concerns sedimentologic and stratigraphic studies of gravelly beaches belonging to Middle Pleistocene terraced marine-deposits outcropping in the vicinity of Bernalda, in an area located between the Bradano and Basento Rivers (Figure 3). Facies analysis has been performed along different sections, where a series of sedimentologic-stratigraphic logs were realized. In accordance with the main methodological manuals relating to sedimentology (i.e., [31,32]) sedimentary facies were distinguished, describing their macroscopic features such as bed thickness, lithology, grain size, and sedimentary structures. The terminology for gravel-sized clasts follows the classification proposed by [33]. Photomosaics were also realized to be useful for following the development of sedimentary bodies and facies, especially in cases of outcrops that are difficult to reach. The location of the measured logs was georeferenced using a GPS. A wide review about the sedimentology of gravelly beaches, briefly reported in the following Section, has been realized in order: (i) to propose a schematic distribution of facies along an idealized depositional profile; (ii) to show constraints about facies distribution/stacking along time during the coastal system development.

## 4. Gravelly Beach

### 4.1. A Brief Sedimentologic Overview

Present-day gravelly coastal systems are made up of coarse-grained particles of various shapes and sizes as a result of redistribution by waves of sediment transported towards the shore by rivers or glaciers or falling down at the foot of seacliffs [34–36]. Basically, gravelly systems show a more inclined slope than sandy ones and are part of reflective beaches rather than dissipative ones (*sensu* [37]).

The depositional profile of a beach system is conventionally divided into three sectors because of its morphology and the hydrodynamics and sedimentary processes that occur [38–40]. Accordingly, proceeding from the submerged coastal area to the exposed one, the three sectors (Figure 5) are represented by: (i) the shoreface, corresponding to the submerged gently sloping coastal belt affected by oscillatory shoaling waves and unidirectional longshore currents; (ii) the beachface, corresponding to the seaward inclined narrow coastal belt straddling the shoreline and affected by surging or just-broken waves that infiltrate or flow down the slope; (iii) the backshore, corresponding to the exposed and most elevated part of the beach affected by seasonal different wave-front action (run-up); it typically shows a sub-horizontal to gently landward-sloping surface.

Commonly, in coarse-grained systems, the beachface and the backshore together are referred to as "the beach", and the transition from beachface to shoreface is often marked by a morphologic step (plunge step) [41–47] (Figure 5). Gravelly beaches in microtidal settings differ from meso- and macro-tidal ones since the beachface, i.e., the sloping sector of the system corresponding to the intertidal zone of "classic" (meso- and macro-tidal) beaches, extends below the low-tide level [48]. Indeed, the term beachface (often indicated as a synonym of foreshore, i.e., the intertidal zone of sedimentary coasts) has acquired a different meaning and has been used by [17] to indicate the whole sloping face of the beach, from the highest berm to the landward boundary of the shoreface (breaker zone). Accordingly, the beachface zone can be subdivided into two subzones: the lower beachface, corresponding to the subtidal part of the slope, and the upper beachface, corresponding to the intertidal part of the same slope (Figure 5).

Mainly following macrotidal examples, gravelly beaches are subdivided into shore-parallel zones (facies belts), with discoidal elements generally deposited in the middle-upper part of the beach (backshore) and spherical-shaped ones accumulated in the lower part (beachface) [48–56]. The gravelly beach zoning is caused by both marine swash and undertow, since these processes are induced by waves whose energy has not been dissipated in the surf zone, a narrow or not-developed zone in gravelly systems [57,58]. Therefore, discoidal clasts are brought up on the backshore, while spherical ones tend to avalanche, rolling down along the slope and accumulating at the foot as a result of the

deceleration of the undertow. Furthermore, since the transport takes place on a gravelly surface that acts as a sieve, the larger elements are transported seaward (overpassing), while the smaller ones become trapped upwards; this process leads to the development of gravelly berms, corresponding to thin wedge-shaped bodies mainly made up of flat clasts.

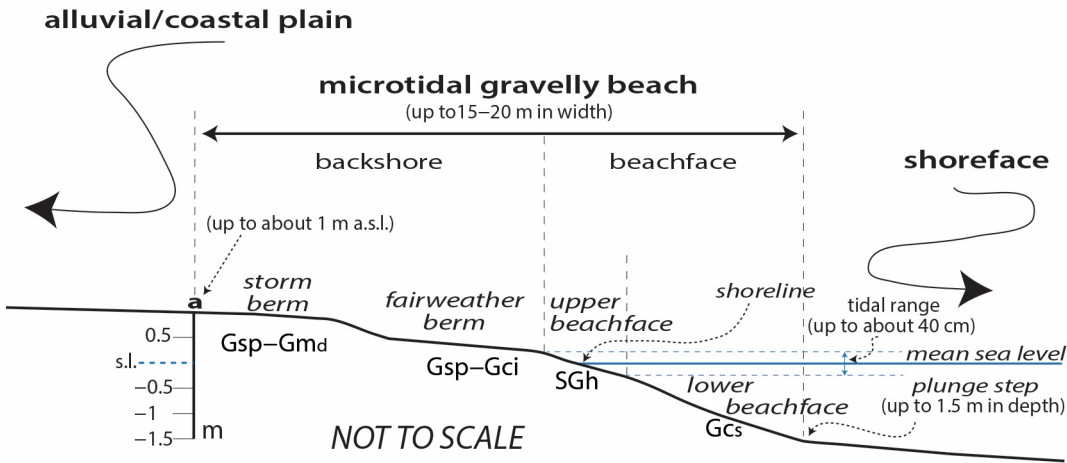

**Figure 5.** Schematic zonation of a coastal system hosting a microtidal gravelly beach. The letter **a** on the left of the profile indicates the maximum landward transport of gravels during storm waves (landward end of gravelly backshore facies); note that the backshore could continue landward, passing to sandy facies, even with a dune system. Three-letter acronyms indicate the facies described in Section 4.1 (Table 1). The reported values refer to microtidal gravelly coastal systems studied in the present-day Mediterranean Sea, according to [59,60] and based on [61].

**Table 1.** Table of facies.

| Facies | Description |
| --- | --- |
| $Gm_d$ | massive gravel with discoidal clasts |
| Gci | clinostratified gravel with imbricated clasts |
| GSp | planar cross—laminated gravel and sand |
| SGh | parallel laminated sand and gravel |
| $Gc_s$ | clinostratified gravel with subspherical clasts |

### 4.2. Why Are Beach Deposits So Important for Constraining Relative Sea-Level Changes?

A very vast body of literature deals with wave-dominated clastic coastal systems (sandy rather than gravelly) and their responses to: (i) the alternation of the seasons; (ii) the variable amount of sediment supply; and (iii) relative sea-level changes.

To start, it is important to highlight that a seasonal beach cycle is the building block of these depositional coastal systems, with the alternation of a swell (fairweather) profile when the beachface grows and a storm profile when the beachface is eroded and the sediment is redistributed to the shoreface [62] (Figure 6a,b). Assuming a stable sea level in a tectonically stable coastal region (no changes in base level) and assuming that input and output of sediment to and from the coast are equivalent, the alternation of beach profiles neither produces a long-term progradation or retrogradation of the coastal system nor its aggradation.

The described "stable" dynamic conditions are unlikely to be realized, and a landward or a seaward shoreline migration can more probably be recorded. During a sea-level still-stand, in clastic coastal systems where sediment supply is greater than sediment transfer to deeper coastal settings, a seaward shoreline migration is recorded. As a consequence, the progradation of the depositional system with the sedimentation of a coarsening and shallowing upward sequence (a regressive sequence) can be observed (i.e., [63]) (Figures 6c and 7a).

By contrast, where seaward sediment transfer exceeds sediment supply, a landward shoreline migration is recorded, accompanied by the erosion of older sediments and the carving of a seacliff (i.e., [64,65]) (Figure 6d).

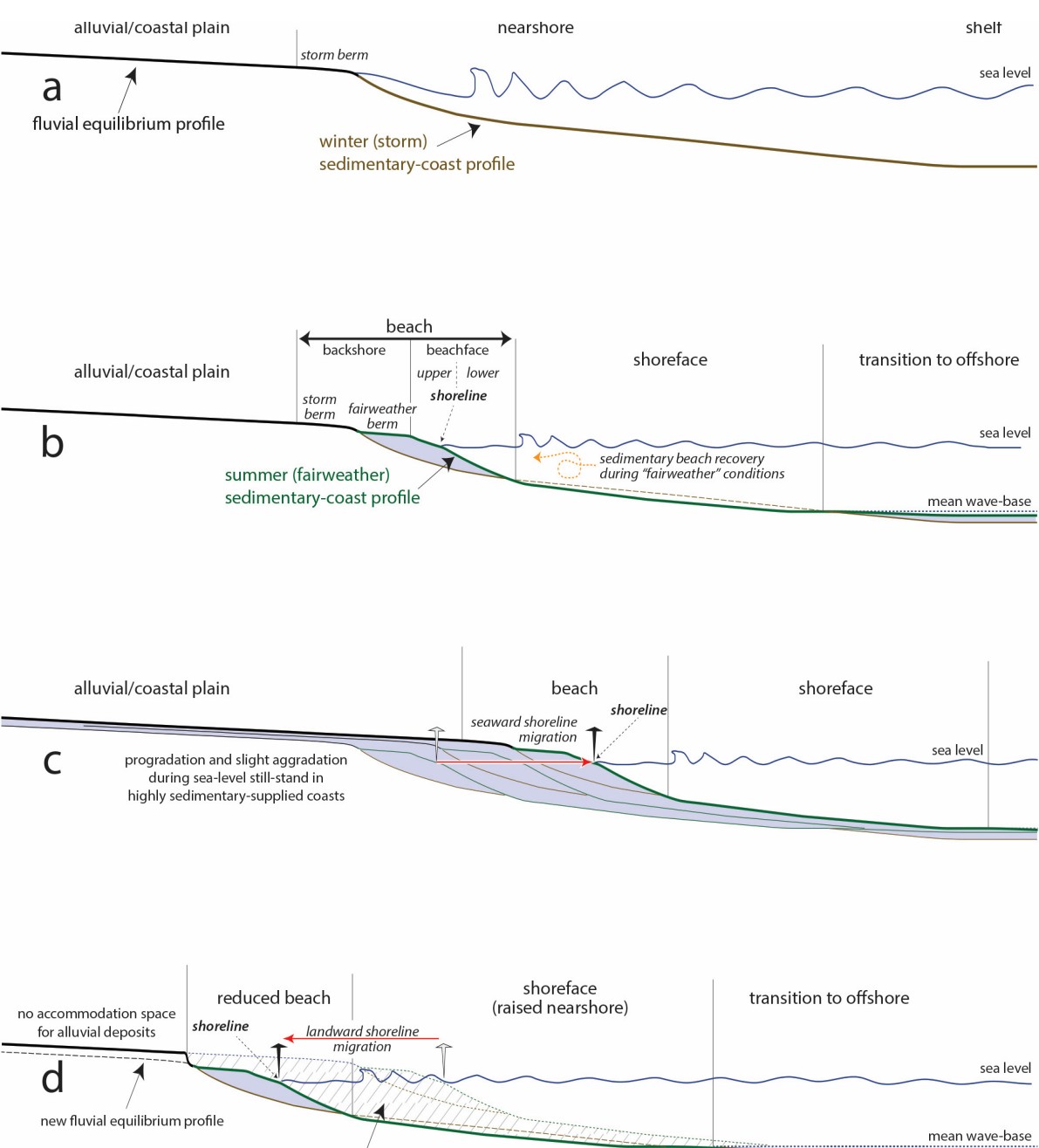

**Figure 6.** Behavior of a beach system during a relative sea-level stillstand. Profiles derive from Figure 5. (**a**,**b**) seasonal response of the beach system to waves. (**c**,**d**) response of the beach system, respectively, in high- and low-supplied sedimentary coasts. Note in (**c**) the small aggradation potential with respect to the amount of progradation potential.

Note that, without change in sea level, even in settings recording high sedimentation rates, the backshore cannot aggrade and the beachface can only prograde [66]. This means that beachface and backshore, even in settings recording variable rates of sediment supply, cannot aggrade without a relative sea-level rise (Figure 7b). Therefore, beaches are the most sensitive environments to relative sea-level changes over time and, during their development, must be considered an excellent proxy of the sea-level position. After [67],

it has been widely accepted that a coastal regressive sequence can be recorded not only during a sea-level stillstand, when, seasonally, sediment inputs exceed sediment outputs (normal regression with constant relative sea level, *sensu* [66]) (Figure 6c), but even during a relative sea-level rise, when the long-term rate of sediment supply exceeds the rate of accommodation space created at the basin margin (normal regression with rising relative sea level, *sensu* [66]) (Figure 8).

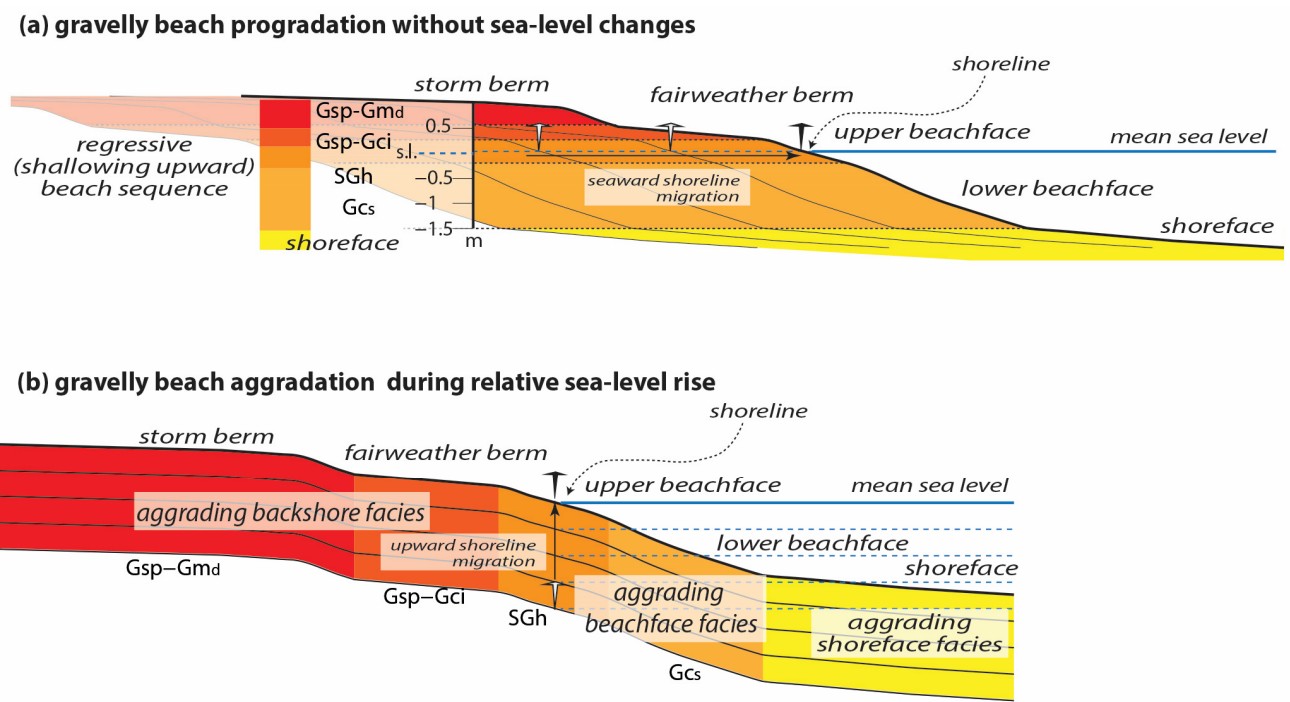

**Figure 7.** (**a**) Seaward shoreline migration of a gravelly beach with a stable sea level (stillstand), in clastic coastal systems where the supply rate is greater than the seaward-transfer rate. The progradation of the beach system allows the sedimentation of a coarsening and shallowing upward sequence (a regressive sequence). Note that there is a little space for the aggradation of backshore deposits (no more than a few tens of centimeters), while the progradation of the system led to the development of relatively thicker beachface deposits, strictly depending on the height of the beachface slope. Compare it with Figure 6c. (**b**) Relative shoreline stationarity during sea-level rise (upward shoreline migration) induces facies aggradation without progradation. A thick vertical stack of the same facies can develop. Profiles derive from Figure 5. Three-letter acronyms indicate the facies described in Section 4.1 (Table 1).

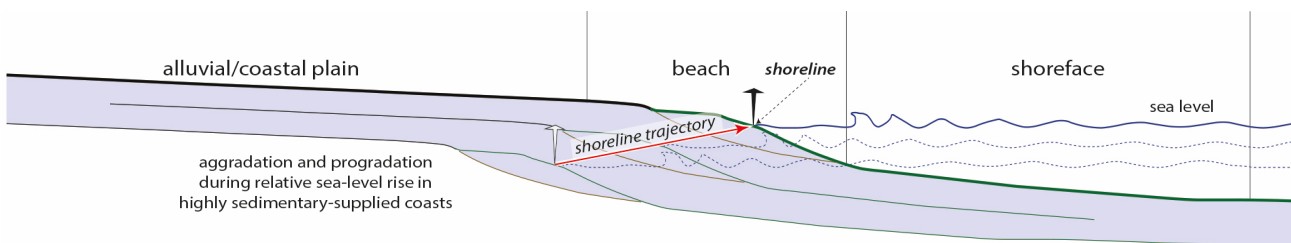

**Figure 8.** Behavior of a beach system during a relative sea-level rise on highly-supplied sedimentary coasts. Note the high potential for aggradation during progradation. Compare with Figures 6c and 7.

As a result of a normal regression with constant sea level, the thickness of the shallowing upward sequence approximates the water depth existing at the beginning of the seaward shoreline migration (Figures 6c and 7a); moreover, the vertical distance of each facies from that of the (upper) beachface represents the original depth (or altitude) of

that environment with respect to the constant relative sea level [68]. Anyway, rarely does this simple scenery develop, and, as described by [69], single beach sequences can be thicker than expected, recording high sediment supply during relative sea-level rise (Figures 7b and 8).

## 5. Facies Analysis

Five facies have been recognized and labeled by a code formed by one or more capital letters, indicating grain size, followed by lowercase letters, indicating main sedimentary structures, and finally by subscript letters, indicating the main shape of clasts (Table 1).

### 5.1. Massive Gravel with Discoidal Clasts ($Gm_d$)

This facies (Figure 9) is made up of massive and clast-supported sub-horizontal up to 20 cm thick gravelly layers, locally with sandy matrix content. The clasts are well-rounded and predominantly discoidal in shape and range in size from pebble to coarse cobble. The layers generally lack sedimentary structures, except for occasional seaward imbrications.

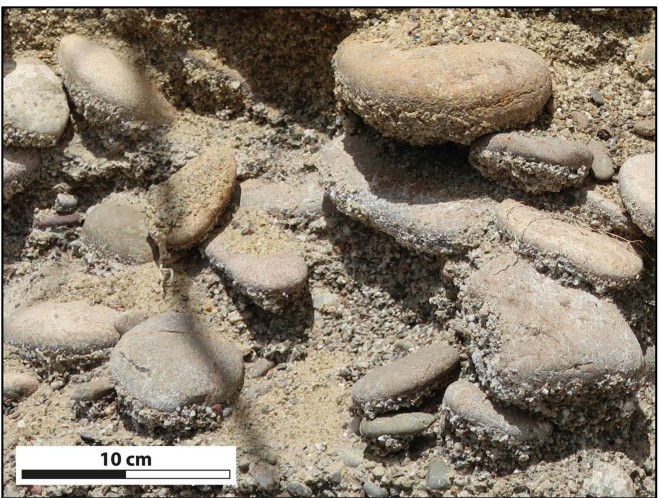

**Figure 9.** Detail of massive gravel with discoidal clasts (facies: $Gm_d$).

Interpretation

Facies $Gm_d$ is the result of storm phases capable of selecting and accumulating discoidal clasts along beaches. It corresponds to the "large-disc zone" *sensu* [49], the "higher berm" *sensu* [17], and the "high storm berm" *sensu* [48].

### 5.2. Clinostratified Gravel with Imbricated Clasts (Gci)

Facies Gci is represented by clinostratified gravelly beds, forming a gently inclined foreset (from 5° to 8°), showing a lenticular geometry. The gravel is basically arranged in well-segregated clast-supported layers with little or no sandy matrix content; layers are often separated by erosive surfaces marked by cobbles. Clasts show mainly flattened shapes, such as discoidal and/or blade-like, and vary in size from pebbles to fine cobbles. The flattened elements also exhibit a well-developed SE-dipping imbrication. Discoidal clasts are mainly found in layers with a thickness ranging from 10 to 20 cm (Figure 10a), while the blade-like ones are mainly found in thin pebbly layers with a thickness generally not exceeding 10 cm (Figure 10b); the former, compared to the previous ones, show a lower degree of selection both for size and shape; in fact, rod-like clasts were found in smaller quantities. Spherical elements are generally scarce and represent a small fraction of the clast content.

Interpretation

This facies is the result of highly selective processes that act preferentially on the shape and size of the clasts and lead to the development of a clear seaward imbrication. Facies of

this type can be linked to traction transport produced by the combination of marine swash and backwash [49,70,71], which leads to the formation of very well-segregated gravelly layers. The presence of numerous erosive surfaces (Figure 11) suggests how selective processes alternate with higher-energy phases, capable of eroding and resedimenting clasts of variable size.

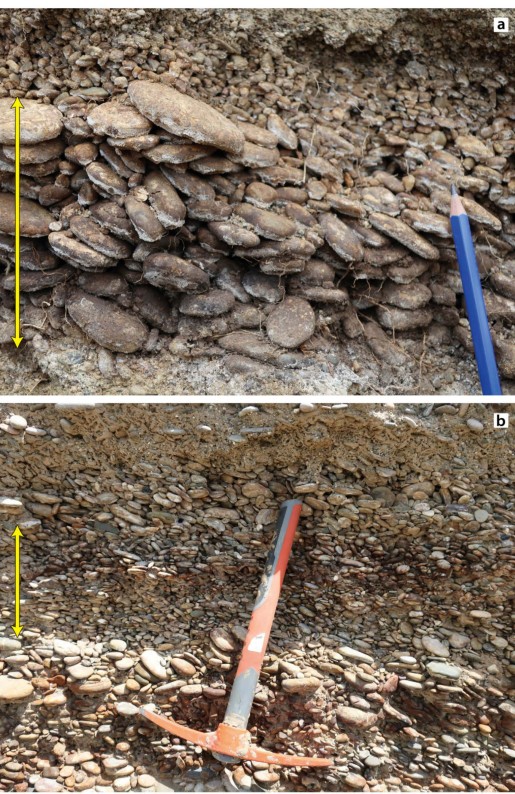

**Figure 10.** Clinostratified gravel with imbricated clasts (facies: Gci) (yellow arrows): (**a**) pebbles to cobbles discoidal clasts exhibiting a well-developed seaward-dipping imbrication (to the right in the photo); (**b**) gravelly layers composed of blade-like imbricated pebbles. The hammer is 30 cm long.

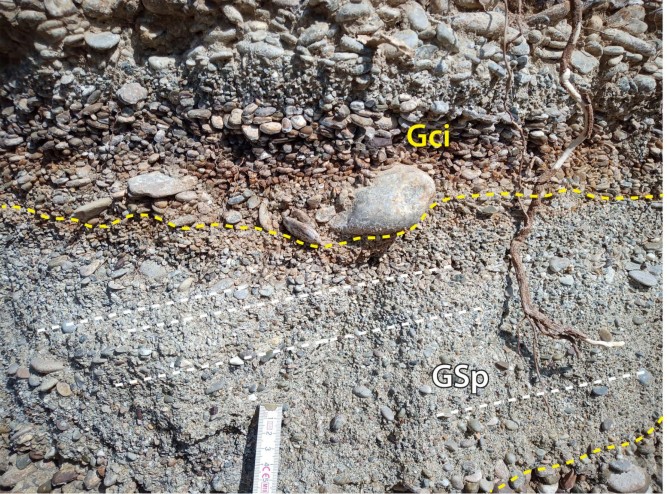

**Figure 11.** From the bottom to the top: planar cross—laminated gravel and sand (facies: GSp) erosively overlain by a cobble-sized lag, in turn passing to clinostratified gravel with imbricated clasts (facies: Gci).

### 5.3. Planar Cross—Laminated Gravel and Sand (GSp)

Facies GSp consists of thin and slightly inclined lenticular gravelly and sandy layers (Figure 11), up to 15 cm thick, with low-angle cross-lamination (not exceeding 12°) mainly dipping landward. The gravelly component consists of well-rounded elements ranging in size from granules to fine pebbles.

Interpretation

Facies GSp can be interpreted as the result of the combined action of marine swash and backwash along the emerging beach. According to [17] and observation of present-day gravelly beaches [60,71], these processes lead to the landward dipping of strata.

### 5.4. Parallel Laminated Sand and Gravel (SGh)

This facies consists of fine- to coarse-grained sandy and gravelly (mainly granules in size) up to 15 cm thick layers, showing a slight inclination with a slope that varies from a few degrees to no more than 6–7°. The examined deposits show a clear parallel lamination (Figure 12).

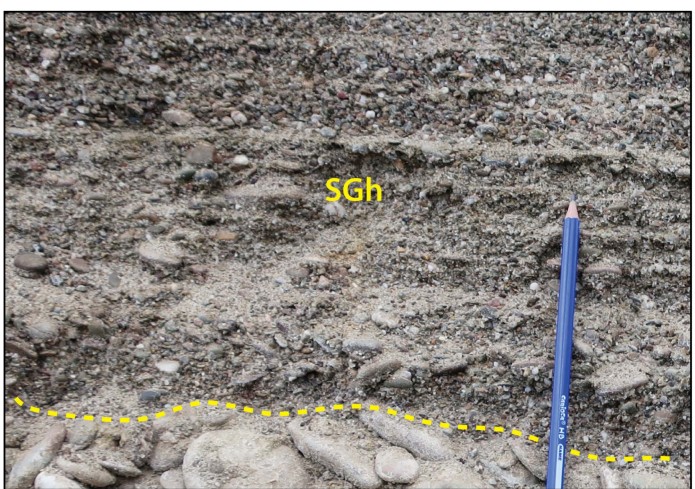

**Figure 12.** Fine- to coarse-grained sand and gravel with parallel lamination (facies: SGh).

Interpretation

The dense plane-parallel lamination allows us to interpret the facies as the result of traction transport by an upper flow regime capable of organizing sands and granules, allowing the development of flat laminae essentially linked to marine swash. The facies is defined by [49] as "sand run".

### 5.5. Clinostratified Gravel with Subspherical Clasts (Gc$_s$)

Facies Gc$_s$ (Figure 13) consists of clast-supported gravelly beds with low to absent sandy–gravelly matrix content. The beds generally range in thickness from 10 to 30 cm and form clinoforms with slopes that can reach 25°. The gravel is composed of well-rounded clasts varying in size from pebbles to coarse cobbles; the latter tend to accumulate downslope in the terminal portions of gravelly layers.

Although several forms are present, subspherical clasts predominate and correspond to the biggest particles, while discoidal, rod, and blade clasts are present in a smaller percentage, the latter two often characterizing the matrix that fills the pores between grains.

Interpretation

Sedimentologic features of the Gc$_s$ facies allow it to be interpreted as the result of mass-transport mechanisms, rheologically comparable to debris fall avalanches [72]. After deposition, the deposits could be partially reworked by wave motion. This gravelly facies

seems to correspond with the "infill zone" of [49], also described in microtidal beach systems by [48].

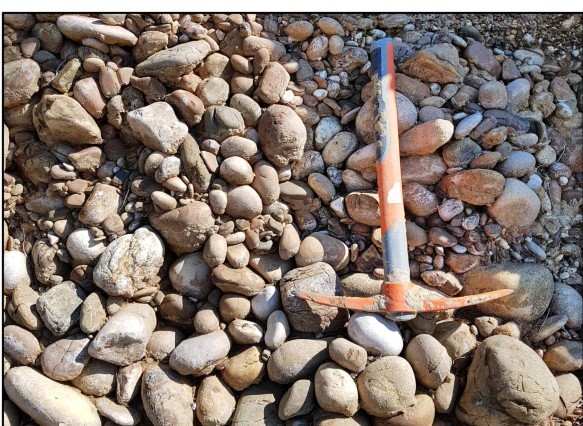

**Figure 13.** Subpherical clasts in the facies Gc$_s$. The hammer is 30 cm long.

## 6. Facies Associations

The five recognized facies were grouped into three facies associations, referring to the following environments: lower beachface, upper beachface, and backshore.

### 6.1. Lower Beachface

The lower beachface is characterized by clinostratified to sigmoidal gravelly bodies made up of up to 30 cm thick seaward SE-dipping layers belonging to Gc$_s$ facies, which accumulate for a total thickness (clinoform height) that rarely reaches 1 m; the superposition of different layers leads to an "apparent" coarsening upward trend, as described by [72]. These deposits pinch out downward, wedging into sandy–gravelly upper shoreface deposits. Seaward dipping of the lower-beachface slope varies from 10° to 25°.

This facies association identifies a beach environment not directly affected by the swash-backwash and located below the intertidal zone (*sensu* [48]), where predominantly subspherical elements accumulate at the foot of the slope thanks to gravity processes triggered in the upper portion of the slope.

### 6.2. Upper Beachface

This facies association consists of gravelly and gravelly–sandy up to 8° clinostratified bodies composed of layers that rarely exceed 20 cm in thickness. This facies association mainly consists of parallel laminated sandy and gravelly layers (SGh) and gravelly layers with imbricated clasts (Gci). The latter exhibit both blade- and rod-shaped pebbles and cobbles with seaward imbrication, as well as spheroidal clasts no larger than pebbles in size. This facies association can be related to beach environments directly affected by marine swash, in particular referable to foreshore environment *sensu* [48], where ephemeral berms could also develop. In the studied microtidal example, this facies association is relatively thin and corresponds to the seaward extent of the swash zone.

### 6.3. Backshore

This facies association consists of sub-horizontal to slightly inclined (no more than 5°), mainly seaward-dipping beds comprising clinostratified gravel with imbricated clasts (Gci), massive gravel with discoidal clasts (Gm$_d$), and sometimes planar cross-laminated gravel and sand (GSp). The facies Gci, which in these deposits completely lacks both sandy matrix and spheroidal clasts, often alternates with the facies GSp; this alternation composes an up to 40 cm-thick vertical stacking of sediments. Facies Gm$_d$ forms gravelly layers no more than 20 cm thick. The main facies features of the described deposits reflect the internal zoning observed in present-day beaches as a result of the highly selective processes

triggered by marine swash and backwash that lead to the development of the main forms found in gravelly beaches, such as ordinary and storm berms [48,49,71].

## 7. Two Examples of Measured Logs and the Record of Sea-Level Changes in Beach Successions

The detailed facies analysis of the studied gravelly beach deposits led to applying the concepts exposed in Section 4.2 to concrete examples. Two of the studied beach sections located in the upper part of two different marine-terraced successions have been selected (Sections 6 and 8 in Figure 3). Section 6 (Spineto-Bernalda locality) shows a beach succession erosively developed onto the foreset of a gravelly delta (Figure 14a). Section 8 (Serra Marina-Bernalda locality) shows the stacking of three gravelly beach successions below the terraced surface (Figure 15). In both cases, linking the genesis of the surface to the whole succession could be wrong.

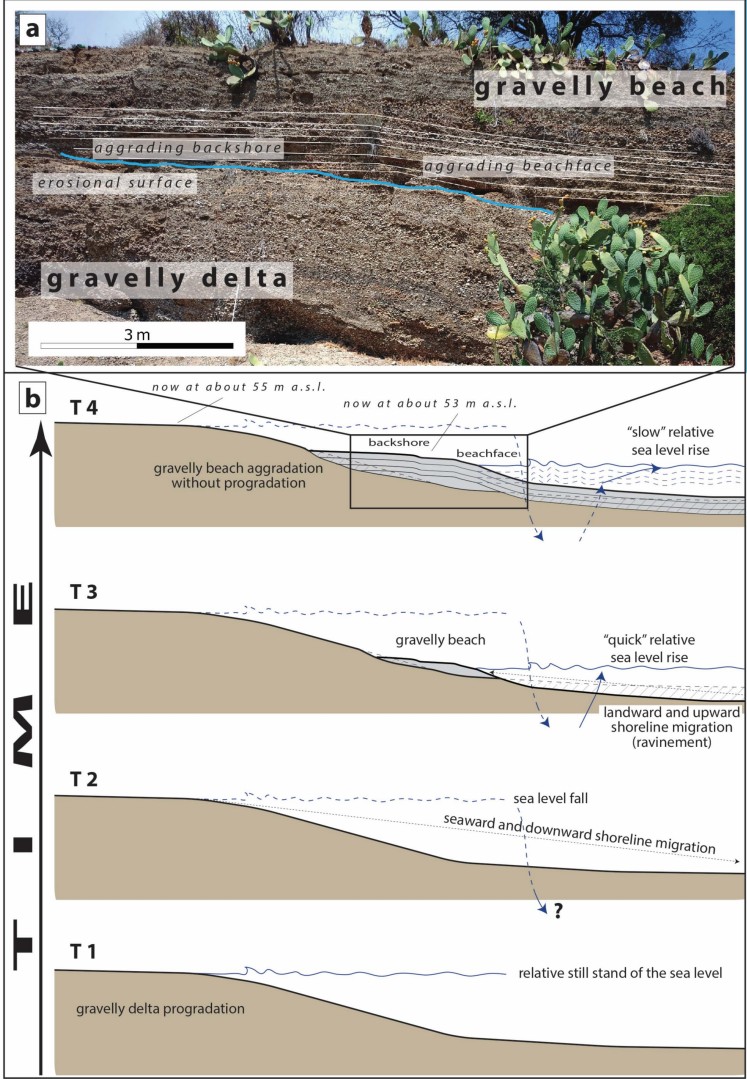

**Figure 14.** (**a**) A gravelly beach succession erosively lying on previous (older) gravelly delta deposits. Spineto locality. See Section 6 in Figure 3. (**b**) To explain the amount of aggrading beach deposits, it is necessary to invoke a relative sea-level rise (compare with Figures 7b and 8) after a relative sea-level fall. To explain the aggradation of backshore facies, it is necessary to invoke the development of a "stationary" shoreline during relative sea-level rise (Figure 7b). The delta must be attributed to a previous (higher) position of the relative sea level. Note the small difference in altitude (about 2 m) between the top of the delta and the topmost layer of the beach. This is an example of a composite terrace, *sensu* [29].

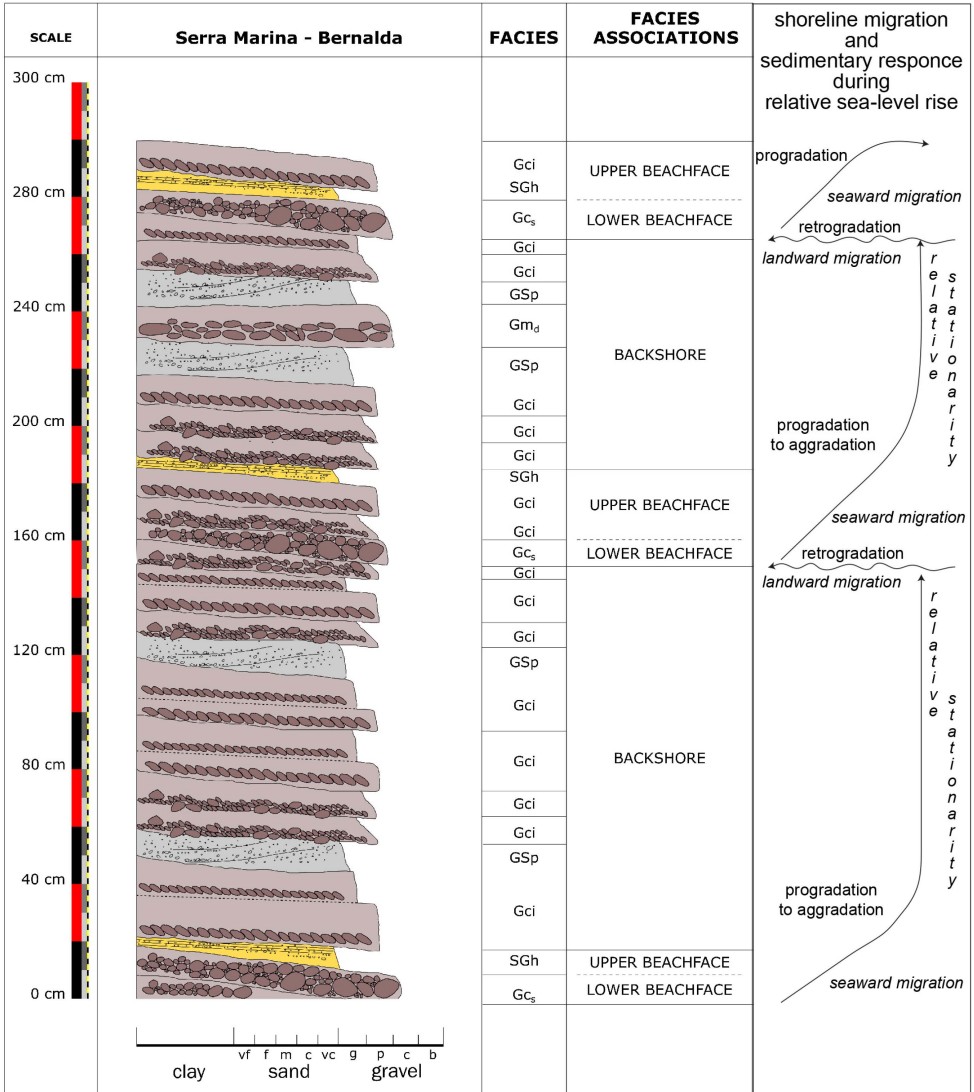

**Figure 15.** Sedimentologic-stratigraphic log performed in Contrada Serra Marina-Bernalda, showing the different facies and facies associations that identify the environments of the gravelly studied beach. Also, shoreline migration and sedimentary response during relative sea-level rise are indicated. See also Figure 7, which explains why backshore facies can aggrade only during a sea-level rise.

### 7.1. Spineto-Bernalda Section

In the locality of Spineto-Bernalda, an along-dip elongated stratigraphic section with respect to the paleo coastline has been studied. The section cuts a succession lying below a topographic surface considered a single marine terrace by all the previous authors. The exposed stratigraphic section shows an aggrading beach succession pinching out on the slope of a gravelly delta (not studied in detail) (Figure 14a). The top of the gravelly delta has an elevation of about 55 m a.s.l. and can be approximated as the record of the relative sea level during the latest phases of the delta progradation (Figure 14b—T1). The topmost layer of the beach succession has an elevation of about 53 m a.s.l. and can be approximated as the record of the maximum relative sea-level rise during the beach aggradation. Since the beach facies lie on an erosional surface, the difference in altitude between the top of the delta deposits and the top of the beach deposits cannot be attributed to a slight sea-level falling during sedimentation. Moreover, the first beach deposits on the erosional surface pertain to backshore environments, and so it is necessary to invoke a relative sea-level fall (whose entity cannot be determined based on outcrop data) followed by a quick rise of the relative sea level to explain the presence of an erosional surface on which backshore

deposits could aggrade (Figure 14—T2 and T3). The first record of the return of the sea on the flank of the delta are backshore deposits, suggesting a sedimentation rate unable to match the creation of new accommodation space (imagine the scheme in Figure 6d applied during a relative sea-level rise). Since the backshore zone is very narrow (Figure 3) and that its facies cannot aggrade too much (no more than a few tens of centimeters—see Figure 7a), the possibility that the related facies could stack for meters is only allowed during a relative stationarity of the shoreline (Figure 14—T4), i.e., if the shoreline lies in the same place (migrate upward) during the rise of the sea level (Figure 7b).

So, the architecture of the outcropping deposits can be explained only assuming that the succession developed during a cycle of relative sea-level change. Therefore, the "terraced surface" of previous authors, i.e., the topographic surface topping both the delta and the beach deposits, cannot be attributed to a single episode of relative sea-level still-stand, but it corresponds to the surface of a composite marine terrace *sensu* [29,30]. Different deposits must be sampled to date the delta top and the beach top. Moreover, the relative sea-level position changed during the beach aggradation, and the possibly dated deposits must be attributed to the sea level rightly correlated to the position of the samples.

*7.2. Serra Marina-Bernalda Section*

In the locality of Serra Marina-Bernalda, an about 3 m thick stratigraphic log was measured and studied in detail (Figure 15). Even in this case, the stacking of backshore deposits should indicate a certain stationarity of the shoreline during a slow relative sea-level rise. Actually, the succession records three episodes of slight drowning of the sedimentary coast (landward migration of the shoreline), since, for three times, lower beachface facies sharply covered backshore ones. This feature records three episodes of moderate transgression that could be explained in two ways:

(a) The sea level was characterized by three slight accelerations in the rate of relative rise, leading to the overlap of relatively deeper environments with shallower ones;

(b) During a constant rate of relative sea-level rise, the supply rate was not constant, and, during time-spans of reduced supply, the beach systems migrated landward. This process could be related to the cyclical switch of a delta mouth located in the vicinity of the studied beach system.

In both cases, the top of the succession indicates the topmost relative rise of the sea level recorded in the locality and is related to the development of that sedimentary coastal system. Even in this case, any dated deposits must be attributed to a relative sea level rightly correlated to the position of the samples, i.e., the top of the succession (the terraced surface) records a sea-level position about 3 m higher than the sea-level position related to the lowermost deposits of the measured log.

## 8. Concluding Remarks

In wave-dominated coastal settings, the most sensitive environment to even the smallest relative sea-level changes is the beachface, which, during its development, also represents the best record (with a small error range) of the relative position of the sea level. Therefore, in order to obtain a better constrained age-dating of uplifted Quaternary successions, beachface recognition represents one of the main tools to correlate deposits to the relative sea level during their formation, especially in microtidal settings.

The Quaternary terraced marine-successions extensively outcropping in the hinterland of Taranto Gulf offer a good opportunity to apply these concepts since gravelly beach deposits abound in the stratigraphic record. The area, besides being an excellent training ground for the study of gravelly beaches in microtidal settings, represents a region where terraced surfaces and marine successions are not yet unanimously attributed to the global curves proposed for the Quaternary sea-level changes.

The reported examples demonstrate that beach deposits: (i) can be used as proxies to highlight even small relative sea-level variations; and (ii) could be the best candidate for dating a terraced surface if the latter is very close and genetically related to sampled sediments.

**Author Contributions:** Conceptualization, L.S. and M.T.; investigation, V.D.G.; writing—original draft preparation, V.D.G., L.S. and M.T.; writing—review and editing, V.D.G., L.S. and M.T.; visualization, V.D.G., L.S. and M.T; supervision, L.S. and M.T.; project administration, M.T.; funding acquisition, M.T. All authors have read and agreed to the published version of the manuscript.

**Funding:** This research was funded by (1) PhD Grant POR Puglia FESR FSE 2014–2020—Asse X—Azione 10.4 "Interventi volti a promuovere la ricerca e per l'istruzione universitaria" Dottorati di ricerca in Puglia XXXIII XXXIV XXXV e XXXVI ciclo (Avviso1/FSE/2020 e 2/FSE/2020) (to V. De Giorgio); (2) PRIN 2022 PNRR—P2022EPCJS: A Warming sea and Coastal retreats around Mediterranean basin—Wa.Co.Med [to S. Andreucci (P.I.) and M. Tropeano (A.I.—RUOR UNIBA)]; (3) "Progetto GeoSciences: un'infrastruttura di ricerca per la Rete Italiana dei Servizi Geologici—GeoSciences IR" (codice identificativo domanda: IR0000037); CUP: I53C22000800006. Piano Nazionale di Ripresa e Resilienza, PNRR, Missione 4, Componente 2, Investimento 3.1, "Fondo per la realizzazione di un sistema integrato di infrastrutture di ricerca e innovazione" finanziato dall'Unione Europea—Next Generation EU (to V. Festa).

**Data Availability Statement:** Not applicable.

**Acknowledgments:** Useful comments and suggestions by D. Ruberti and the other two anonymous reviewers helped us improve the manuscript. Exposed data and interpretations are part of the PhD Project "Sedimentological study of paralic coarse-grained Pleistocene deposits on top of Bradanic Trough successions between the Bradano and Basento Rivers (Basilicata, Southern Italy)" realized at the University of Bari Aldo Moro (Italy) by V. De Giorgio under the supervision of L. Sabato and M. Tropeano.

**Conflicts of Interest:** The authors declare no conflict of interest.

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
