# Peer review of "Gravelly Beach Deposits as a Proxy for Relative Sea-Level Changes in Microtidal Wave-Dominated Shoreline Systems: Examples from the Hinterland of the Taranto Gulf (Middle Pleistocene, Basilicata, Southern Italy)"

_water, doi:10.3390/w15203631_

Round 1

Reviewer 1 Report

The manuscript provides an extensive and detailed description of gravelly beach coastal systems in a microtidal environment. The concepts are well documented and the references provide a solid background for reference. Specific comments are reported in the attached file.

I would suggest reporting the names of the authors cited in the text where specific texts or concepts are referred to, such as "sensu..." or "according to...", in this way: sensu Tizio et al. [25]. In fig. 3, in particular, to help the reader link the authors mentioned in the figure to the reference list, it is preferable to recall them, in the caption, next to the reference number.

It would also be helpful in reading the facies to provide a summary table of the main facies described and their interpretation in terms of facies association.

In the reference list, check that the publication date is in bold throughout. Add the number of pages to references n. 3, 4, 15, 19, 20, 21, 50, 51.

Please, check the text for typos and english

The English of the text is very clear and correct. There are many typos for which it is suggested to carefully check all text.

Reviewer 2 Report

General comments

Dear Editor, 

I hereby report my comments to the paper “Gravelly beach deposits as a proxy for relative sea-level changes in microtidal wave-dominated shoreline systems: examples from the hinterland of the Taranto Gulf (Middle Pleistocene, Basilicata, Southern Italy)” by Di Giorgio V., Sabato L. and Tropeano M.  (ID: water-2565364).

The paper deals with stratigraphy and sedimentology of coastal terraced deposits. The topic is of general interest and deserves publication on Water. The methodology is traditional but correct. The paper is well written and description is well separated from the interpretation. I claim for minor changes before publication. These are summarized as follows:

 1.       The title is very long I would shorten it to: “Gravelly beach deposits as a proxy for relative sea-level changes in microtidal wave-dominated shoreline systems: examples from the Taranto Gulf (Middle Pleistocene, Southern Italy)”

2.       Section 2 would benefit of a schematic figure (also from literature) summarizing the zonation of gravelly beaches in relation to waves and tides.

3.       The geological setting should provide more information about the geodynamic setting (subsidence/uplift rates, orogenic processes, position of the study area relative to the orogen). These data are crucial for the definition of one of the main components of relative sea level at time of formation of the described deposit: tectonic movements.

4.       Section 7.1, in my opinion does not respond to the question in the title: Why are beach deposits so important for constraining relative sea-level changes? In fact, the section does not clarify why beach deposits are better than other sea-level indicators. To me, in this part of the paper the author highlights the fact that beach deposits, differently from other sea-level indicators, can provide clues on the relationships between relative-sea-level oscillation and changes in sediment budget of the coastal system. Therefore, I suggest to change the title.

5.       In section 7.3 the authors present new data, so this part should be moved to the Results after section 6. In general, whereas the facies analysis is very detailed, this stratigraphic part, with only two examples, is a bit weak.

6.       The discussion could be improved with more reference to the geodynamic setting (see comment 3) and to published global sea-level curve. Although age attribution of terraces and underlying deposits is not unequivocal, without a reference to the Pleistocene eustatic and tectonic history the reader is a bit lost.

 7.       Throughout the manuscript there is a bit of confusion between depositional environment (e.g. delta) and associated deposits (e.g. delta gravels). Please see minor comments below.

Minor comments and suggestions:

 Line 23: the exposed successions

Line 24: the Quaternary

Line 41: I suggest to change “collocate” to “place”

Line 46: across the Bradanic Trough

Line 51: to low-amplitude sea-level oscillations

Line 52: which represents one of the best indicators of palaeo sea-level

Line 53: to develop a future

Line 62: by a terraced surface

Lines 58-60: how did you pursue these aims? Add a sentence about methods.

Line 67: redistribution by waves of sediments

Line 70: delete of particles, or change “grain size” to “particle size”.

Lines 69-72: Split the sentence in two shorter ones.

Line 73: The depositional profile of what?

Line 74: which area?

Line 84-86: Join this paragraph to the previous.

Line 87: change “examples” to “settings” and “were” to “have been”.

Line 96: the smaller ones

Line 97: change “allows” to “leads”.

Line 99: I suggest to delete “Example of”

Line 106: delete “of the zone”

Line 111: the Southern Apennines foredeep

Lines 111-112: Middle and Upper Pleistocene

Line 112 and 116: delete “-“

Line 115: Argille Subappenine Fm.

Line 115: change “followed” to “overlain”.

Line 116: delete “and younger”

Line 117: down to the Metaponto Coastal Plain

Line 117: delete coarse-grained

Line 118: marine or coastal?

Lines 118-119: change “where it is possible to distinguish either” to “and include”.

Line 121-123: add “(see Fig. 2 for location”).

Lines 126-128: Based on detailed facies analysis of tens of coarse-grained successions, exposed close to the study area (or within the study area?), locally exceeding 40 m in thickness.

Lines 128-129: a composite ideal vertical suite of shallow-marine to alluvial facies, showing a regressive trend.

Lines 131-132: these sedimentary successions topped by terraced surfaces record more than a single relative sea-level oscillation.

Line 134: study

Lines 135-136: in a … km2-wide area between Basento and Bradano Rivers (Figs. 2 and 3).

Lines 139-140: please add references

Lines 146-147: you can delete this first sentence

Line 158: I would say subspherical

Line 160: delete “associated with”.

Line 162: I would change “of these deposits” to “between grains”

Line 165: delete “a”

Line 171: change “sometimes” with “locally”

Line 172: range

Line 184: 10-20 cm-thick

Line 185: marked by cobble-layers

Line 187: you can delete “they”

Line 197: can be linked to? due to?

Line 213: overlain by

Lines 216-217: please explain how the process act. Why clinobeds are inclined landward? Or quote some reference paper

Line 222: are laminae given by subtle changes in grainsize. It seems to me that laminae are marked by thin cemented sandy layers. Am I correct? If yes, you could add this observation to your description.

Line 232: The five recognized facies were grouped into three facies associations…

Lines 232-239: Join the two paragraphs in a single one

Line 235: These environments…

Lines 237-239: Do these data derive from your facies analysis or from literature? In the second case, please add references.

Line 243: change “overlapping” to “superposition”. What do you mean by “apparent”?

Line 244: deposits are not inclined. Strata or laminae are inclined.

Line 245: what does “those” refer to? to the inclination? To the deposits?

Line 245: the genetically related deltaic feeding area, or a modern deltaic feeding area? I suppose you refer to the genetically related deltas. In this case in lines 244-249, there is no clear separation between description and interpretation. Indeed, you call into question a deltaic environment which you reconstructed based on interpretation of sediment bodies. If you want to refer to deltaic deposits you should describe and interpret the associated deposits.

Lines 255-256: This description need to be re-organized: e.g. “This facies association consists of gravelly and gravelly-sandy bodies, composed of decimetric layers which rarely exceed 20 cm in thickness. This facies association mainly consists of horizontal laminated sandy and gravelly layers (SGh) and up to 8°clinostratified gravelly layers with imbricated clasts (Gci).” In this way you avoid repetitions and you assign an inclination to the layer and not to the sediment bodies.

Line 270: change “producing” to “composing”.

Line 271: change “repetition” to “stacking”

Line 275: delete “of”

Line 278: I would change “wide” to “vast” and delete “mainly”

Line 280: I would add climate change and autogenic processes (e.g. delta-lobe switch)

Line 281: a seasonal beach cycle is the building block of these coastal successions

Line 286: I would delete source and sink which are not exactly equivalent to input and output.

Line 289: I suggest to change “very difficult to realize” with “unlikely”

Line 290: can be more probably recorded (delete “more easily”)

Line 290: During a sea-level still-stand

Line 291: where sediment supply is greater than sediment transfer to the deep sea

Line 292: given by the progradation…

Lines 290-295: this sentence is too long and the cause-effect relationships are not very clear (see previous comment). I suggest to rephrase is and possibly to split into two shorter sentences.

Line 295: By contrast

Lines 295-298: where sediment transferred to the deep sea exceeds sediment supply.

Lines 299-301: I don’t think this is the reason why beaches are excellent sea-level indicators (see Rovere et al., 2016)

Line 318: the number of surface and the age

Line 321: without entering

Line 338: I would delete “ones”

Line 344: I would change “constrained to” to “framed into”

Line 357: please say something about the dating method

Line 362-364: that dropped and led to the formation of an erosive surface. Deposition of gravelly beach deposits took place during the following relative sea-level rise and highstand.

Lines 368-369: Rephrase. Eg. “Section 6 shows a beach sequence deposited on top of the sloping erosive surface of delta gravels (or delta gravel deposits)”. Here it’s not clear if the sloping surface at the top of delta graves is a depositional or erosive.

Line 371: stratigraphic section 6, elongated…

Line 372: cuts

Line 373: beneath a terraced surface

Lines 373-374: the exposed succession

Line 374: of gravelly delta deposits

Line 382: cannot be determined based on outcrop data.

Line 382: are there data to say “quick rise”? Moreover, how fast? I would simply say “a relative-sea-level rise”

Line 384-387: An interesting aspect is the absence of a paleosol on top of delta gravels. This fact suggests deposition soon after the development of the erosive surface without a prolonged period of subaerial exposure. In this case, the deposition of beach deposits on top of delta gravels could simply reflect a delta-lobe switch in a contest of general subsidence. Therefore, the invoked relative sea-level cycle could have been recorded only locally due to a intrinsic process of the delta-coastal system. Please discuss this point.

Fig. 2: the figure is a geological map where colours represent geological units. To me “marine terraces” is a geomorphological feature. Therefore, in the legend I would change “marine terraces” to “marine terraced deposits”.

Fig. 3: STUDIED GRAVELLY BEACH DEPOSITS. Bradano and Basento Rivers in Italic. “River” with initial capital letter.

Fig. 12: A. gravelly beach deposits, backshore deposits, beachface deposits, delta deposits. B. “without progradation”: if stratal surfaces are inclined, how is it possible that there is no progradation

I'm not an English mother tongue, but the English language sounds to me ok, with only few corrections (see minor comments)

Reviewer 3 Report

SEE REMARKS IN THE ATTTACHMENT

The overalll concept is valauble, but parts of the ms need urgent redressing.

Round 2

Reviewer 3 Report

The  revised ms is significantly improved. The added figures and tables are helpful.

A few added sentences should be checked.